# Genetic Analysis of Milk Production Traits and Mid-Infrared Spectra in Chinese Holstein Population

**DOI:** 10.3390/ani10010139

**Published:** 2020-01-15

**Authors:** Chao Du, Liangkang Nan, Lei Yan, Qiuyue Bu, Xiaoli Ren, Zhen Zhang, Ahmed Sabek, Shujun Zhang

**Affiliations:** 1Key Lab of Agricultural Animal Genetics, Breeding and Reproduction of Ministry of Education, Huazhong Agricultural University, Wuhan 430070, China; dc1992hml@163.com (C.D.); 15827557518@163.com (L.N.); ahmedsabek1987@gmail.com (A.S.); 2Henan Dairy Herd Improvement Center, Zhengzhou 450000, China; yanleihcy@163.com (L.Y.); buqiuyue103@163.com (Q.B.); renxl1990@163.com (X.R.); zzgux@163.com (Z.Z.); 3Department of Veterinary Hygiene and Management, Faculty of Veterinary Medicine, Benha University, Moshtohor 13736, Egypt

**Keywords:** mid-infrared spectra, milk production traits, spectral wavenumbers, heritability, genetic correlation

## Abstract

**Simple Summary:**

Usually, spectral data are used as predictors to predict milk components, animal characteristics, and even reproductive status. Another innovative way to use spectral data involves considering spectral wavenumbers as traits and then analyzing from the genetic perspective. In this study, we considered milk spectral data directly as traits, then detected the influence of some non-genetic factors on spectral wavenumbers and estimated the genetic parameters of spectral points. The result of the present study could be used as a management tool for dairy farm and also provides a further understanding of genetic background of milk mid-infrared (MIR) spectra. In future, milk spectral data could be applied more effective. For example, some sub-clinical diseases might be detected based on the difference between the expected and observed values of the spectral traits. In addition, we could also use genetic correlation between wavenumbers and a trait of interest, which are difficult and expensive to measure, to apply for the genetic improvement of dairy species.

**Abstract:**

Milk composition always serves as an indicator for the cow’s health status and body condition. Some non-genetic factors such as parity, days in milk (DIM), and calving season, which obviously affect milk performance, therefore, need to be considered in dairy farm management. However, only a few milk compositions are used in the current animal selection programs. The mid-infrared (MIR) spectroscopy can reflect the global composition of milk, but this information is currently underused. The objectives of this study were to detect the effect of some non-genetic factors on milk production traits as well as 1060 individual spectral points covering from 925.92 cm^−1^ to 5011.54 cm^−1^, estimate heritabilities of milk production traits and MIR spectral wavenumbers, and explore the genetic correlations between milk production traits and 1060 individual spectral points in a Chinese Holstein population. The mixed models procedure of SAS software was used to test the non-genetic factors. Single-trait animal models were used to estimate heritabilities and bivariate animal models were used to estimate genetic correlations using the package of ASReml in R software. The results showed that herd, parity, calving season, and lactation stage had significant effects on the percentages of protein and lactose, whereas herd and lactation stage had significant effects on fat percentage. Moreover, the herd showed a significant effect on all of the 1060 individual wavenumbers, whereas lactation stage, parity, and calving season had significant effect on most of the wavenumbers of the lactose-region (925 cm^−1^ to 1200 cm^−1^), protein-region (1240 cm^−1^ to 1600 cm^−1^), and fat-regions (1680 cm^−1^ to 1770 cm^−1^ and 2800 cm^−1^ to 3015 cm^−1^). The estimated heritabilities for protein percentage (PP), fat percentage (FP), and lactose percentage (LP) were 0.08, 0.05, and 0.09, respectively. Further, the milk spectrum was heritable but low for most individual points. Heritabilities of 1060 individual spectral points were 0.04 on average, ranging from 0 to 0.11. In particular, heritabilities for wavenumbers of spectral regions related to water absorption were very low and even null, and heritabilities for wavenumbers of specific MIR regions associated with fat-I, fat-II, protein, and lactose were 0.04, 0.06, 0.05, and 0.06 on average, respectively. The genetic correlations between PP and FP, PP and LP, FP, and LP were 0.78, −0.29, and −0.14, respectively. In addition, PP, FP, and LP shared the similar patterns of genetic correlations with the spectral wavenumbers. The genetic correlations between milk production traits and spectral regions related to important milk components varied from weak to very strong (0.01 to 0.94, and −0.01 to −0.96). The current study could be used as a management tool for dairy farms and also provides a further understanding of the genetic background of milk MIR spectra.

## 1. Introduction

It is known that milk composition changes when a cow has a metabolic disorder such as acidosis, ketosis, or mastitis. Therefore, milk composition always serves as an indicator for the cow’s health status and body condition. In order to better manage dairy farm and enhance productivity of cows, it is substantial to develop an understanding of the factors affecting milk performance. Some non-genetic factors such as parity, days in milk (DIM), and calving season, which obviously affect milk performance, therefore, need to be considered for sustaining breed improvement.

It is also known that milk is a very complex product consisting of many components. However, only a few milk components (for example, fat, protein, and lactose percentages) are included in the current animal selection programs. The reasons for this are that analytical methods such as gas chromatography or HPLC to quantify milk components are expensive and time-consuming, and these methods are less suitable for large-scale routine measurement in the dairy farm. Mid-infrared (MIR) spectroscopy is a widely used spectroscopic technique based on crossing matter by electromagnetic radiation and on the subsequent measure of energy absorption in the mid-infrared region [1,2]. As a rapid, accurate, and cheap technique, MIR spectroscopy has been implemented as an effective tool in the prediction of milk phenotypes [1], such as the contents of protein [3], fat [4], fatty acids [5], and major minerals [6]. The spectral data can reflect the global composition of milk, but this information is currently underused. Therefore, it could be interesting to study directly the spectral information for management and breeding. A study on the non-genetic factors affecting milk spectral data is therefore justifiable.

It is demonstrated that milk spectra also have the potential to predict animal characteristics, such as methane emissions [7], energy balance [8], and feed intake [9]. Attempts have been also made to assess the reproductive status of dairy cows, such as oestrus [10,11] and pregnancy stage [12,13] from milk spectra. Milk spectra are used as predictors to predict milk phenotypes, animal characteristics, and even reproductive status as mentioned above. Other innovative ways to use spectral data also exist, that is, milk spectral data is considered as traits and then be analyzed by genetic models [12]. Actually, genetic analyses of milk spectra have been revealed by several previous studies [14,15,16,17], represented by various heritabilities ranging from low to high. However, a knowledge gap still exists regarding the genetic basis of milk spectra. The genetic correlations of milk spectral data to traits of interesting have not been reported before. 

Therefore, the specific aims of the present study were to (1) estimate the effects of non-genetic factors on milk production traits and 1060 individual spectral points covering MIR region from 925.92 cm^−1^ to 5011.54 cm^−1^, (2) calculate heritabilities (h^2^) of milk production traits and spectral points, and (3) calculate genetic correlations between milk production traits and spectral points. 

## 2. Materials and Methods

### 2.1. Original Data

Original data were provided by the Henan Dairy Herd Improvement Center. The original data including information on cow parity, calving date, days in milk (DIM), milk production traits such as protein percentage (PP), fat percentage (FP) and lactose percentage (LP) predicted by mid-infrared (MIR) spectroscopy, and spectral data were collected from August to October 2018 in 41 herds and included a total number of 9954 records from 8694 cows. The contents of milk protein, fat, lactose, as well as spectral information containing 1060 infrared transmittance points in the region between 925.92 cm^−1^ to 5011.54 cm^−1^ were determined by CombiFoss FT+ (Foss, Hillerød, Denmark). In addition, there were 9516 animals including 8206 females and 1310 males in the original pedigree file. Each animal was traced back as many generations as possible.

### 2.2. Data Editing

The outliers of the milk production traits including the contents of milk protein, fat, and lactose were defined as samples whose values deviating more than 3 standard deviations from the mean of the corresponding trait. Based on this definition, the numbers of observations available for PP were 9893 from 8645 cows, for FP were 9871 from 8626 cows, and for LP were 9804 from 8592 cows. 

Spectral points expressed in transmittance (T) were transformed to absorbance (A) with the equation A = log_10_(1/T). Principal component analysis carrying out on the edited spectral data showed that the first 9 principal components explained more than 90% of total spectral variation. Therefore, robust Mahalanobis distance was performed from the first 9 principal component scores to detect spectral outliers, and removal of spectral outliers was based on 97.5% quantile of the chi-squared distribution with 9 degrees of freedom. Finally, 9322 spectral records from 8200 cows were retained. 

A new data set was then used for genetic correlations analyses between milk production traits and 1060 individual spectral points. This new data set consisting of 9292 records from 8175 cows was obtained through removal both the outliers of milk production traits and spectral outliers from the original data set.

### 2.3. Models for Estimates of Non-Genetic Factors

The mixed models procedure of SAS software was used to test the effects of non-genetic factors. Calving season, parity, lactation stage, and herd on PP, FP, LP, and 1060 individual spectral points were tested. In this study, calving seasons were classified into 4 groups based on climatic conditions of the area, including March to May as spring, June to August as summer, September to November as autumn, and December to February as winter. Parities of cows were grouped into 3 levels, namely, 1st, 2nd, and ≥3rd. Lactation stages were divided into 5 levels, including 1–50 days, 51–100 days, 101–200 days, 201–305 days, and >305 days. Therefore, the model was: Y_ijklm_ = μ + S_i_ + P_j_ + L_k_ + H_l_ + cow_m_ + e_ijklm_, where Y_ijklm_ was the response variable; μ was overall mean for each trait; S_i_ was the fixed effect of calving season (i = 1, 2, 3, 4); P_j_ was the fixed effect of parity (j = 1, 2, 3); L_k_ was the fixed effect of lactation stage (k = 1, 2, 3, 4, 5); H_l_ was the fixed effect of herd (l = 1, 2, 3, …, 41); cow_m_ was the random effect of cows accounting for the repeated measurements; and e_ijklm_ was the random error. Bonfferoni correction was used to adjust for multiple testing of the differences between effect levels, and *p* < 0.05 was considered significant.

### 2.4. Models for Genetic Analyses

Heritabilities of milk production traits and spectral points were estimated using single-trait animal model REML method by the package of ASReml in the R software. The model was: y = Xb + Z_1_a + Z_2_p+e, where y was a vector of observations; b was a vector of fixed effects; a was a vector of random animal additive effects; p was a vector of random permanent environmental effects; e was a vector of random residual effects; X, Z_1_ and Z_2_ were incidence matrixes assigning observations to effects. Genetic correlations between milk production traits and spectral points were estimated using the bivariate animal models also by the package of ASReml in the R software. Only significant (*p* < 0.05) effects of fixed factors on milk production traits and 1060 individual spectral points were used in the genetic analyses. In addition, the pedigree data consisted of 9468 animals including 8161 females and 1307 males in the final data set of PP; 9450 animals including 8143 females and 1307 males for FP; 9414 animals including 8113 females and 1301 males for LP; 9038 animals including 7758 females and 1280 males for spectral data; and 9015 animals including 7736 females and 1279 males for genetic correlations analyses between milk production traits and 1060 individual spectral points.

## 3. Results and Discussion

### 3.1. Descriptive Statistics 

The descriptive statistics of three milk production traits were presented in Table 1. Similar to data reported by Coffey et al. [18] and Mele et al. [19], the average of PP, FP, and LP was 3.45%, 4.19%, and 4.91%, with coefficients of variation of 0.13, 0.27, and 0.05, respectively (Table 1). 

### 3.2. Estimates of Fixed Effects on Milk Production Traits 

All fixed factors including herd, parity, calving season, and lactation stage had significant effects on PP and LP, whereas only herd and lactation stage had significant effects on FP. The estimates of parity, calving season, and lactation stage effects on milk production traits are shown as follows. 

#### 3.2.1. Parity

The estimates of parity effect on milk production traits were presented in Figure 1. Our results revealed that the parity had a significant (*p* < 0.05) effect on PP. The PP was significantly greater in the second parity (3.41%) than in the first (3.37%) and the third parity (3.38%) (Figure 1A). Bovenhuis et al. [20] reported that protein percent increased from parity 1 to 2, and was slightly lower in parity 3 as compared with parity 2. Zhao et al. [21] also observed the highest protein content in the second parity. These results were in accordance with the present study. The effect of parity on fat percentage was negligible. The FP in the first and second parities was slightly higher than that in the third parity (Figure 1B). Zhao et al. [21] observed that fat content in parity 2 was slightly higher than that in parity 1 and 3. The LP significantly decreased with increasing parity, declining, on average, by 0.13% from parity 1 to parity 3 (Figure 1C). Bovenhuis et al. [20] reported that lactose percentage decreased from parity 1 to 2 and, subsequently, to parity 3, which was in line with the results of the present study. The same decreased tendency of lactose content with increasing parity was also observed by Haile-Mariam et al. [22]. 

#### 3.2.2. Calving Season

The effect of calving season on milk production traits were presented in Table 2. The results showed that calving season had a significant (*p* < 0.05) effect on PP and LP. The highest PP was observed in the winter (3.43%) and the least PP was observed in summer (3.34%). The low milk protein percentage of cows calving in summer can be explained mainly with the depression effect of high temperatures. In addition, as is well known that the phenotypic correlation between PP and LP is always negative [23]. Therefore, it was reasonable that the lactose percentage was the highest in summer (4.95%) and the lowest in the winter (4.92%), which was on the contrary of PP. There was no significant difference of fat content among different calving seasons, which was in agreement with Verma et al. [24], who also reported that there was no significant effect of calving season on milk fat percentage, which was the result from a small population (259 Sahiwal cattle with 600 records). 

#### 3.2.3. Lactation Stage 

The estimates of the different lactation stages on milk production traits were shown in Table 3. Lactation stage showed significant (*p* < 0.05) effect on PP, FP and LP. Both PP and FP were the highest in the fifth milking stage (3.68% and 4.28%) and the lowest in the second lactation stage (3.17% and 3.77%). Results for the effect of lactation stage on protein and fat percentages in the current study were in agreement with findings of Bovenhuis et al. [20] and Weller et al. [25], who observed that, during the entire lactation stage, protein and fat contents decreased from the beginning of the lactation to about 50 DIM, and then increased until the ending of the lactation. On the contrary, LP was the highest in the second milking stage (5.01%) and the lowest in the fifth lactation stage (4.83%), which was supported by Bovenhuis et al. [20], who described the increase of lactose percentage from 0 DIM to 50 DIM, and subsequently, the decrease from 50 DIM to 390 DIM. 

### 3.3. Estimates of Fixed Effects on Milk Infrared Spectra

Herd showed a significant effect on all of the 1060 individual wavenumbers, including wavenumbers of the water absorption regions (1600 cm^−1^ to 1700 cm^−1^, and 3040 cm^−1^ to 3660 cm^−1^), which might reflect feeding and management differences between farms. Nowadays, consumers have increasing interest in purchasing organic milk. The authenticity of milk get more and more noticed. Therefore, infrared spectroscopy could be one of the methods that enable discriminate whether milk produced by cows from organic farms or not. Lactation stage had significant effect on all of the wavenumbers of the lactose-region (925 cm^−1^ to 1200 cm^−1^), protein-region (1240 cm^−1^ to 1600 cm^−1^), and fat-regions (1680 cm^−1^ to 1770 cm^−1^ and 2800 cm^−1^ to 3015 cm^−1^), while most of the wavenumbers of these specific regions related to important milk components were significantly affected by parity and calving season. It is well known that milk components are influenced by non-genetic factors such as parity, lactation stage, and calving season, which is also demonstrated in the present manuscript. Therefore, the effects of these fixed factors on milk components are reflected in the forms of the milk infrared wavenumbers in the current study. Further, most of the wavenumbers of the water absorption regions (1600 cm^−1^ to 1700 cm^−1^, and 3040 cm^−1^ to 3660 cm^−1^) were not significantly affected by parity, calving season, and lactation stage. However, there were a few wavenumbers of the water absorption regions that were significantly affected by these fixed factors. For instance, wavenumbers 3040 cm^−1^ to 3090 cm^−1^ were significantly affected by both parity and lactation stage, whereas wavenumbers 3553 cm^−1^ to 3657 cm^−1^ were significantly affected by lactation stage. The wavenumbers of the water absorption region are usually excluded when setting up prediction models. Our results indicate that the wavenumbers of water absorption regions that were significantly affected by these fixed factors may contain some important information on milk components. 

### 3.4. Heritability

The estimated heritabilities of PP, FP, and LP were 0.08, 0.05, and 0.09, respectively. Actually, heritabilities of PP, FP, and LP were various among different previous studies, ranging from low to high. Fleming et al. [26] calculated that the h^2^ for PP and FP were 0.52 and 0.51, respectively. Buitenhuis et al. [27] presented the h^2^ for PP was 0.47 in Holstein cows and 0.70 in Jersey cattle, respectively. Costa et al. [28] reported that heritability of LP was 0.43. The estimated h^2^ of PP, FP, and LP in the present study were much lower compared with the above previous studies but similar to Zhang et al. [29], who estimated that the heritabilities of PP and FP were 0.08 and 0.07, respectively. The discrepancies across studies regarding the heritabilities for milk production traits could be due to several factors, such as differences in total records, number of records of per animal, data editing, breeds, the statistical model, and the method of calculating heritability. For example, Fleming et al. [26] estimated the genetic parameters using multiple-trait random regression test-day models by Bayesian methods based on 49,127 test-day milk samples from 10,029 first parity Holstein cows in 810 herds, while in the current study, we calculated heritability using single-trait animal model based on only 9322 records from 8200 Chinese Holstein cows instead of only a single test-day record of per cow. Further, the very short period (only three months) of data collection and the very limited pedigree used may be another reason for low heritabilities of milk production traits in the present study.

Figure 2 presented heritabilities for 1060 individual wavenumbers of the spectral region from 925.92 cm^−1^ to 5011.54 cm^−1^. Heritabilities of the 1060 wavenumbers were 0.04 on average, ranging from 0 to 0.11. In total, 167 spectral points yielded heritability estimates that were lower than 0.01 and even null, which were corresponding to two high noise level spectral regions related to water absorption (1600 cm^−1^ to 1700 cm^−1^, and 3040 cm^−1^ to 3660 cm^−1^). Three hundred and ninety-five spectral points exhibited heritability estimates between 0.01 and 0.04, whereas 328 and 132 spectral points had heritability in the intervals of 0.04 to 0.07 and 0.07 to 0.10, respectively, and 38 spectral points had heritability greater than 0.1 (Table 4). In addition, the estimated average heritabilities for wavenumbers of the specific regions including lactose- region (925 cm^−1^ to 1200 cm^−1^), protein-region (1240 cm^−1^ to 1600 cm^−1^), and fat-regions (1680 cm^−1^ to 1770 cm^−1^ and 2800 cm^−1^ to 3015 cm^−1^) were 0.06, 0.05, 0.04, and 0.06, respectively (Table 5). The heritability estimation of milk spectra has been revealed by several previous studies. Bittante and Cecchinato [15] indicated that the milk spectrum was heritable for most individual wavenumbers. Heritabilities for wavenumbers of milk spectra ranged from 0.003 to 0.42 in Holstein cows [14], from 0 to 0.63 in Holstein Friesian cows [16], from 0 to 0.31 in Danish Holstein, and from 0 to 0.3 in Danish Jersey [17]. Previous studies [15,17] and the present study showed the similar pattern of heritabilities for the different wavenumbers. The heritabilities for wavenumbers of the two regions related to water absorption was null or very low. Meanwhile, most of the comparatively high-heritability wavenumbers were associated with important milk components. However, similar to the milk production traits, h^2^ of individual wavenumbers in the current study was also considerably lower comparing with the previous studies. Besides the reasons mentioned above, the possible reasons also include different breeds, sample size and instruments used. For example, Zaalberg et al. [17] estimated heritabilities for milk infrared wavenumbers that were from 3275 Danish Holstein cows with 19,656 records and from 3408 Danish Jersey cows with 20,228 records, comparing with this study which was from 8200 Chinese Holstein cows with only 9322 records. Further, Bittante and Cecchinato [15] estimated heritabilities for milk spectrum using the MilkoScan FT120 FTIR interferometer containing 1056 spectral points, comparing with the present study based on CombiFoss FT+ containing 1060 spectral points. However, the most likely reasons are the analytical method and statistical models applied. For instance, estimation of h^2^ by Zaalberg et al. [17] followed a Bayesian approach. In the study of Soyeurt et al. [14], authors first used principal components analysis to reduce the dimensionality of the spectra, and estimated variances on the transformed scales were then back transformed towards the original 1060 spectral traits using a 2-step approach. Variance components for the spectral points were estimated by a single-trait REML animal model in the studies of Bittante and Cecchinato [15], Wang et al. [16] as well as the present study. However, comparing with the study of Bittante and Cecchinato [15], the lower heritability in the present study may be that, besides fixed factors of herd, DIM, and parity, random effect of animal additive genetic effect, we considered the calving season as a fixed factor, and added permanent environmental effect in the statistical models. In addition, the classification of the same fixed factors, including herd and DIM had a little difference between these two studies, representing by 30 herds vs. 41 herds and 10 levels of days in milk vs. five levels of lactation stage. 

### 3.5. Genetic Correlation 

The genetic correlation between PP and FP was 0.78 in this study. Fleming et al. [26] also detected a strong genetic correlation between PP and FP (0.72), which was in line with the present study. Further, the genetic correlations between PP and LP, FP and LP were −0.29 and −0.14, respectively, which was similar to Senddon et al. [30], who reported that the genetic correlations of LP with PP and LP were −0.14 and −0.05, respectively. 

In this study, we also demonstrated the genetic correlations between milk production traits and the spectral wavenumbers. As shown in Figure 3, PP, FP, and LP shared the similar patterns of genetic correlation with the spectral wavenumbers. 

Within the lactose-region (925 cm^−1^ to 1200 cm^−1^), wavenumbers 1180 cm^−1^ and 925 cm^−1^ showed the strongest positive (r_g_ = 0.78) and negative (r_g_ = −0.85) genetic correlation with fat percentage, and the average positive and negative genetic correlation between spectral points from the lactose-region and fat percentage was 0.43 and −0.60, respectively. Wavenumbers 1199 cm^−1^ and 929 cm^−1^ showed the highest positive (r_g_ = 0.71) and negative (r_g_ = −0.70) genetic correlation with protein percentage, and the average positive and negative genetic correlation between spectral points from the lactose-region and protein percentage was 0.40 and −0.43, respectively. Almost all of the spectral points within the lactose-region were positively correlated with lactose percentage, and the average positive genetic correlation was 0.44. Further, wavenumber 1026 cm^−1^ had the highest positive genetic correlation (r_g_ = 0.73) with lactose percentage (Table 6). 

Within the protein-region (1240 cm^−1^ to 1600 cm^−1^), wavenumbers 1242 cm^−1^ and 1597 cm^−1^ showed the highest positive (0.76) and negative (−0.71) correlations with fat percentage, and the average positive and negative genetic correlation between wavenumbers of protein-region and fat percentage were 0.32 and −0.28, respectively. Most of the spectral points within the protein-region were positively correlated with protein percentage, and the average positive genetic correlation between spectral points within the protein-region and protein percentage was 0.47. Further, wavenumber 1539 cm^−1^ had the highest positive genetic correlation (r_g_ = 0.94) with protein percentage. The negative genetic correlation between spectral points within the protein-region and lactose percentage was low and even negligible. Wavenumber 1539 cm^−1^ has the highest positive genetic correlation (r_g_ = 0.70) with LP, and the average positive genetic correlation between spectral points within the protein-region and lactose percentage was 0.44 (Table 6). 

Within the first fat-region (1680 cm^−1^ to 1770 cm^−1^), wavenumbers 1736 cm^−1^, 1743 cm^−1^, and 1724 cm^−1^ showed the highest positive correlations with FP, PP, and LP (r_g_ = 0.78, 0.57, and 0.67, respectively). The average positive correlations between spectral points within the first fat-region and FP, PP, and FP were 0.74, 0.50, and 0.13, respectively. Meanwhile, wavenumbers 1720 cm^−1^, 1724 cm^−1^, and 1682 cm^−1^ had the highest negative correlations with FP, PP, and LP (r_g_ = −0.88, −0.96, and −0.15, respectively). The average negative correlation between spectral points within the first fat-region and FP, PP, and LP were −0.67, −0.57, and −0.06, respectively. Within the second fat-region (2800 cm^−1^ to 3015 cm^−1^), all spectral points showed the positive genetic correlation with lactose percentage. Wavenumber 2827 cm^−1^ had the highest correlation (r_g_ = 0.47) with LP, and the average r_g_ between wavenumbers within the second fat-region and lactose percentage was 0.21. Wavenumbers 2947 cm^−1^ (r_g_ = 0.80) and 2966 cm^−1^ (r_g_ = 0.80) had the highest positive genetic correlation with FP and PP, respectively. The average positive r_g_ between wavenumbers within the second fat-region and FP and PP were 0.65 and 0.62, respectively. Wavenumbers 3013 cm^−1^ had the highest negative genetic correlation with both FP (r_g_ = −0.69) and PP (r_g_ = −0.53). The average negative r_g_ between wavenumbers within the second fat-region and FP and PP were −0.54 and −0.42, respectively. 

## 4. Conclusions

The present study shows that non-genetic factors such as herd, parity, calving season, and lactation stage have significant effects on milk protein and lactose percentages, whereas herd and lactation stage have significant effects on milk fat percentage. Furthermore, herd shows significant effect on all of the 1060 individual wavenumbers, while lactation stage has significant effect on all of the wavenumbers of the lactose-region (925 cm^−1^ to 1200 cm^−1^), protein-region (1240 cm^−1^ to 1600 cm^−1^), and fat-regions (1680 cm^−1^ to 1770 cm^−1^ and 2800 cm^−1^ to 3015 cm^−1^), and most of the wavenumbers of these specific regions related to important milk components are significantly affected by parity and calving season. In addition, we confirms that heritabilities for wavemunbers of spectral regions related to water absorption are very low or even null, and the percentages of milk fat, protein, and lactose, as well as most of the spectral wavenumbers associated with important milk components are heritable but low. The percentages of milk protein, fat, and lactose share the similar patterns of genetic correlation with the spectral wavenumbers. The current study can be used as a management tool for dairy farms and also provides a further understanding of genetic background of milk MIR spectra.

## Figures and Tables

**Figure 1 animals-10-00139-f001:**
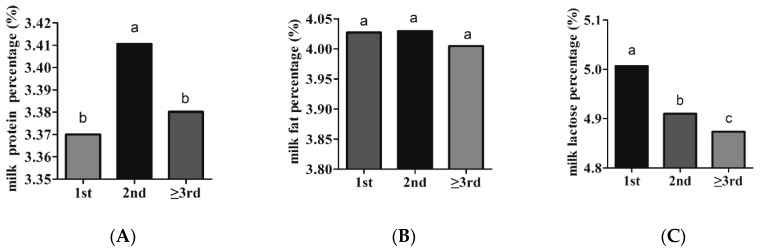
Estimates of the parity effects on milk production traits, including (**A**) protein percentage, (**B**) fat percentage, and (**C**) lactose percentage.

**Figure 2 animals-10-00139-f002:**
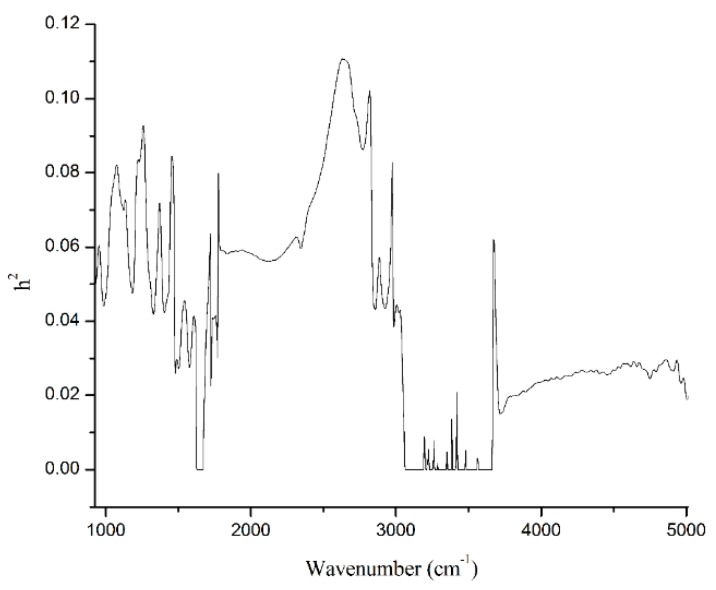
Heritabilities (h^2^) for 1060 individual spectral points of infrared region from 925.92 cm^−1^ to 5011.54 cm^−1^.

**Figure 3 animals-10-00139-f003:**
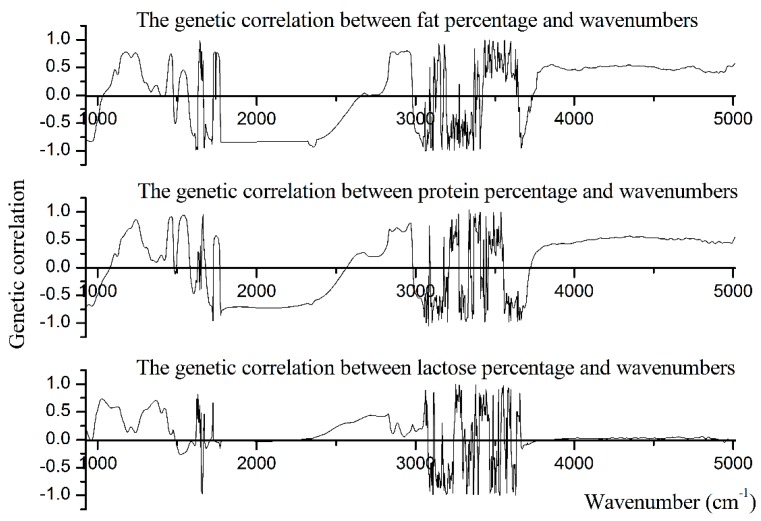
The genetic correlations between milk production traits and wavenumbers of spectral region from 925.92 cm^−1^ to 5011.542 cm^−1^.

**Table 1 animals-10-00139-t001:** Descriptive statistics of milk production traits.

Traits	Records	Mean	SD	CV	Minimum	Maximum
PP	9893	3.45	0.44	0.13	2.06	4.92
FP	9871	4.19	1.13	0.27	0.70	7.84
LP	9804	4.91	0.26	0.05	3.94	5.62

PP = protein percentage (%); FP = fat percentage (%); LP = lactose percentage (%).

**Table 2 animals-10-00139-t002:** Estimates of the calving season effects on milk production traits.

Season of Calving	PP		FP		LP	
	*n*	Mean ± SE	*n*	Mean ± SE	*n*	Mean ± SE
Spring	2029	3.37 ± 0.01 ^b,c^	2024	3.97 ± 0.03 ^a^	2017	4.93 ± 0.01 ^a,b^
Summer	2198	3.34 ± 0.01 ^c^	2183	4.01 ± 0.03 ^a^	2178	4.95 ± 0.01 ^a^
Autumn	2405	3.41 ± 0.01 ^a,b^	2404	4.04 ± 0.03 ^a^	2375	4.93 ± 0.01 ^b^
Winter	3261	3.43 ± 0.01 ^a^	3260	4.06 ± 0.03 ^a^	3234	4.92 ± 0.01 ^b^

PP = protein percentage (%); FP = fat percentage (%); LP = lactose percentage (%). ^a–c^ within a row, means sharing different superscripts differ significantly (*p* < 0.05).

**Table 3 animals-10-00139-t003:** Estimates of the lactation stage effects on milk production traits.

Lactation Stage	PP		FP		LP	
	*n*	Mean ± SE	*n*	Mean ± SE	*n*	Mean ± SE
1–50 days	1204	3.20 ± 0.01 ^d^	1194	4.06 ± 0.04 ^b^	1203	4.96 ± 0.01 ^b^
51–100 days	1018	3.17 ± 0.02 ^d^	1010	3.77 ± 0.04 ^d^	1012	5.01 ± 0.01 ^a^
101–200 days	1613	3.36 ± 0.01 ^c^	1608	3.88 ± 0.03 ^c^	1606	4.96 ± 0.01 ^b^
201–305 days	3500	3.52 ± 0.01 ^b^	3487	4.11 ± 0.03 ^b^	3473	4.90 ± 0.01 ^c^
>305 days	2558	3.68 ± 0.01 ^a^	2572	4.28 ± 0.03 ^a^	2510	4.83 ± 0.01 ^d^

PP = protein percentage (%); FP = fat percentage (%); LP = lactose percentage (%). ^a–d^ within a row, means sharing different superscripts differ significantly (*p* < 0.05).

**Table 4 animals-10-00139-t004:** Estimates of heritabilities (h^2^) for 1060 spectral points, according to the value of heritability.

h^2^	Spectral Points Number	Heritability			
		Average	SD	Minimum	Maximum
<0.01	167	0.00046	0.0015	0	0.009
0.01–0.04	395	0.026	0.0045	0.01	0.04
0.04–0.07	328	0.055	0.0073	0.04	0.07
0.07–0.1	132	0.083	0.0084	0.07	0.1
>0.1	38	0.107	0.0037	0.1	0.11
All	1060	0.041	0.029	0	0.11

**Table 5 animals-10-00139-t005:** Estimates of heritabilities (h^2^) for 1060 spectral points, according to MIR regions.

MIR Region		Heritability			
Name	Range (cm^−1^)	Average	SD	Minimum	Maximum
Lactose	925–1200	0.06	0.01	0.04	0.08
Protein	1240–1600	0.05	0.02	0.03	0.09
Fat-I	1680–1770	0.04	0.01	0.02	0.06
Fat-II	2800–3015	0.06	0.02	0.04	0.10
Total	925–5011	0.04	0.03	0.00	0.11

**Table 6 animals-10-00139-t006:** Overview of genetic correlations between milk production traits and MIR regions.

Traits	MIR Regions (cm^−1^)	Negative Correlation				Positive Correlation			
		Average	SD	Min	Max	Average	SD	Min	Max
PP	925–1200 (lactose-region)	−0.43	0.24	−0.02	−0.70	0.40	0.22	0.01	0.71
	1240–1600 (protein-region)	−0.22	0.14	−0.06	−0.42	0.47	0.32	0.01	0.94
	1680–1770 (fat-region)	−0.57	0.23	−0.10	−0.96	0.50	0.09	0.32	0.57
	2800–3015 (fat-region)	−0.42	0.12	−0.17	−0.53	0.62	0.15	0.11	0.80
FP	925–1200 (lactose-region)	−0.60	0.28	−0.02	−0.85	0.43	0.28	0.02	0.78
	1240–1600 (protein-region)	−0.28	0.25	−0.02	−0.71	0.32	0.22	0.02	0.76
	1680–1770 (fat-region)	−0.67	0.21	−0.07	−0.88	0.74	0.04	0.63	0.78
	2800–3015 (fat-region)	−0.54	0.19	−0.11	−0.69	0.65	0.22	0.10	0.80
LP	925–1200 (lactose-region)	−0.01	0	−0.01	−0.02	0.44	0.23	0.01	0.73
	1240–1600 (protein-region)	−0.17	0.09	−0.03	−0.26	0.44	0.21	0.01	0.70
	1680–1770 (fat-region)	−0.06	0.05	−0.01	−0.15	0.13	0.27	0.01	0.67
	2800–3015 (fat-region)	/	/	/	/	0.21	0.12	0.06	0.47

PP = protein percentage (%); FP = fat percentage (%); LP = lactose percentage (%).

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
