# Peer review of "Genetic Analysis of Milk Production Traits and Mid-Infrared Spectra in Chinese Holstein Population"

_animals, 2020, doi:10.3390/ani10010139_

Round 1

Reviewer 1 Report

Respected Author,

I was reading your submission with high interest.

Even the topic is not totally new, you used existing data (information on wawenumbers) in new way.

Even methodically and mathematically, sound estimated heritabilities are weak as:

1) in comparison to estimation based on regular phenotypic data from animal recording and are extremelly low

2) there are relativelly high standard deviation in case of estimated heritabilities and correlations

Threfore I would expect more disscussion and clear statement in the conclusions explaining those facts. Otherwise, conclusion that the estimates and methodology are perspective in further breeding is in doubt.

Reviewer 2 Report

The manuscript deals with the estimation of genetic parameters of milk quality characteristics. Including a very simple approach under animal model. And analysis of a new set of features obtained through MID infrared analysis.

My concerns:

There is a need for greater and better quality discussion of the results obtained, especially about new findings, which may be potentially useful. There is also a need for better presentation of the real advantage of using the new features rather than the traditional mode in which they are used for the prediction of the fatty acid profile and other components of the milk profile.

Reviewer 3 Report

GENERAL COMMENTS

The manuscript is interesting and exploits several applications of Mid-infrared Spectra under genetic and breeding approaches in dairy cattle. However, several issues must be elucidated as reported below.

SIMPLE SUMMARY

Lines 16-18: “Another innovative ways to use spectral data is that spectral wavenumbers are considered as traits and then are analyzed from the genetic perspective”

Please, see lines 58-60 in the “Introduction” section. “In addition, milk spectral points can be considered as traits and can then be analyzed by genetic models [12]. Actually, genetic analyses of milk spectra have been revealed by several previous studies [14-17], represented by various heritabilities ranging from low to high.”

What are the mentioned “innovative ways” in comparison with previous studies listed at References [12] and [14-17]?

Lines 18-19: “This study opens up a new insight to further understanding of genetic background of milk spectra.”

What are the mentioned “new insights” in comparison with previous studies listed at References [12] and [14-17]?

ABSTRACT

Lines 29-30: “Single-trait animal models were used to estimate variance components using the R software”

How “to explore the genetic correlations” reported as a aim of this study under a Single-trait animal model framework? For this, multiple-trait models are required. More explanations are needed in this point.

Lines 30-33: Currently, there are no reasons to omit uncertainty measures (such as the standard error) for heritability estimates. This kind of information is required here.

Line 35: “were similar to those estimated for these traits.”

How similar were they? What about to report summary statistics (such as average correlation) in order to validate this information in Abstract section?

Lines 35-36: “The genetic correlations 35 between PP and FP, PP and LP, FP and LP were 0.78, -0.29 and -0.14, respectively.”

Firstly, as mentioned before, how to calculate genetic correlations using single trait models? Secondly, what about standard errors for these estimates?

Lines 36-38: “In addition, we are the first to demonstrate the genetic correlations between milk production traits and the spectral wavenumbers in this study.”

Please, see comments about “new insights” at lines 16-19 in the “simple sample” section.

Lines 40-42: “were disorganized, and the genetic correlations between milk production traits and spectral regions related to important milk components varied from weak to very strong.”

The terms “disorganized” and “weak to very strong” must be “translated” into numbers. For example, what means “disorganized” under an animal breeding approach? What are the correlation values to be defines as “weak” and “very strong”?

Lines 42-44: “The current study confirms the heritable of the milk spectral wavenumbers and provides a further understanding of genetic background of milk MIR spectra.”

There is no scientific support (numerical information and statistical uncertainty measures) in the Abstract section to conclude on “heritable of the milk spectral wavenumbers”. Please, this conclusion must be revised. Additionally, the abbreviation “MIR” was not previously defined in this section.

INTRODUCTION

Lines 61-63: “For example, the effects of non-genetic factors, such as parity and days in milk (DIM) on the milk spectra, and the genetic correlations between milk production traits and spectral wavenumbers have not yet been reported.”

What are the advantages to treat “spectral wavenumbers” directly as traits? This explanation is the basis of the present manuscript and must be better exploited here.

Lines 64-65: “(1) estimate the effects of non-genetic factors on milk production traits and 1,060 individual spectral points covering”

This aim was not previously reported and discussed in the “Abstract section”. This aim must be revised.

Lines 67-69: “The present study provides a further understanding of the genetic background of milk spectra and also offers a new insight to use milk spectral data more effective and innovative in future.”

These features were not “proven” based on the summarized results presented in the Abstract section. I think that this information would me revised before to be included here.

MATERIALS AND METHODS

Lines 78-80: “In addition, there were 9,516 animals including 8,206 females and 1,310 males in the original pedigree 79 file.”

How many generations were considered in this pedigree? What is the average inbreeding of this population? This kind of information is often used in dairy breeding, and would be presented here.

Lines 90-91: “removal of spectral outliers was based on 97.5% quantile of the chi-squared distribution with 9 degrees of freedom.”

Why 97.5% quantile of the chi-squared distribution? What is the justification (based on previous reference or statistical properties) to use this threshold? Additionally, the degrees of freedom would be revised, since the number of variables is exactly 9. Please, more information is required here.

Lines 93-94: “A new data set was then used for genetic correlations analyses between milk production traits and 1,060 individual spectral points.”

Why the dataset was changed to realize “genetic correlations analyses”? It is not common (because it implies in loss of information) in genetic analysis based on multiple-trait mixed models. The justifications are not clear and would be presented here.

Line 98: “The mixed models procedure of SAS software was used to test the effects of non-genetic factors.”

What about ANOVA properties such as residuals normality for “spectral wavenumbers”? Have the authors tested the normality for a trait that is defined as a counting data (wavenumbers)? The GLIMMIX procedure of SAS software is often recommend for this kind of analysis. Some comments are required here.

Line 104: The notation of the presented model must be rewritten using the indexes “ijklm” as “subscript”.

Lines 107-108: “cow was the random effect of cows 107 accounting for the repeated measurements”

If the authors exploited “repeated measurements” approach, more information about the tested and used covariance matrix (VC, CS, CSH, AR1, etc…) of PROC MIXED must be better explained.

Lines 108-109: “was the random error which was assumed to be randomly and independent distributed, with zero and constant variance.”

How to use a “repeated measurements” model assuming that residuals are “independent distributed”? What was the used covariance matrix (VC, CS, CSH, AR1, etc…) giving support to the mentioned assumption? This information must be better explained.

Lines 113-114: “Heritabilities of milk production traits and spectral points were estimated using single-trait animal model REML method in the R software.”

What was the used package of “R software”? This information is vague because there are several R packages to fit mixed models under an animal breeding viewpoint.  More detais are required here.

Lines 117-118: “Genetic correlations between milk production traits and spectral points were estimated using the bivariate animal models.”

This is very important information that was omitted from previous section. For example, at the Abstract section the authors reported only the use of single trait models. Additionally, what was the R package used to fit a multiple-trait model? Finally, what was the algorithm implemented under REML? EM, PXEM, AI, etc… What was the convergence criterion applied to ensure the convergence of all fitted models? This kind of information is essential and must be presented here.

RESULTS AND DISCUSSION

This section was not defined in this manuscript. Thus, the results with respective discussions were considered together within “material and methods” section. The manuscript must be revised in terms of sections organization.

Lines 128-194: All this results and discussion are related to “Estimates of Fixed Effects on Milk Production Traits”. This aim was not exploited in the Abstract section. There is no information about all this content in Abstract. Additionally, there are no reports about the relevance of studying “non-genetic factors” in the “Introduction” section. The lack of information about these results in previous sections reduces the quality of the present manuscript.

Tables 4, 5 and 6. Standard errors are required in order to provide uncertainty measures for heritability and genetic correlation estimates reported in the present manuscript.

 Main question (again…): What are the advantages to treat “spectral wavenumbers” directly as traits? This explanation is the basis of the present manuscript and must be better exploited here. There are few “explanations” about these “advantages” under a breeding program framework. More information about how to exploit “spectral wavenumbers” directly as traits can be attractive for breeders in the field of Animal Science and Genetics.  

CONCLUSIONS

Lines 317-318: “Furthermore, we are the first to demonstrate the genetic 317 correlations between milk production traits and the spectral wavenumbers in this study”

I don’t know if this “phrase” can be interpreted as a conclusion. Please, it must be revised.

Lines 318-321: Please, see the definitions previously “criticized” in other section:  “The terms “disorganized” and “weak to very strong” must be “translated” into numbers. For example, what means “disorganized” under an animal breeding approach? What are the correlation values to be defines as “weak” and “very strong”?”

Lines 322-326: All this content presented here do not represent “real conclusions” based on results obtained in the present study.

Round 2

Reviewer 3 Report

All comments, corrections and suggestions were considered in this revised version. 

This manuscript is a resubmission of an earlier submission. The following is a list of the peer review reports and author responses from that submission.